# Classification Algorithms Used in Predicting Glaucoma Progression

**DOI:** 10.3390/healthcare10101831

**Published:** 2022-09-22

**Authors:** Filip Tarcoveanu, Florin Leon, Silvia Curteanu, Dorin Chiselita, Camelia Margareta Bogdanici, Nicoleta Anton

**Affiliations:** 1Ophthalmology Department, Faculty of Medicine, University of Medicine and Pharmacy “Gr. T. Popa” Iasi, University Street No 16, 700115 Iasi, Romania; 2Faculty of Automatic Control and Computer Engineering, “Gheorghe Asachi” Technical University of Iasi, 27 Mangeron Street, 700050 Iasi, Romania; 3Department of Chemical Engineering, Faculty of Chemical Engineering and Environmental Protection “Cristofor Simionescu”, “Gheorghe Asachi” Technical University of Iasi, 73 Mangeron Street, 700050 Iasi, Romania

**Keywords:** glaucoma, machine learning, artificial intelligence, classification algorithms, predictions

## Abstract

In this paper, various machine learning algorithms were used in order to predict the evolution of open-angle glaucoma (POAG). The datasets were built containing clinical observations and objective measurements made at the Countess of Chester Hospital in the UK and at the “St. Spiridon” Hospital of Iași, Romania. Using these datasets, different classification problems were proposed. The evaluation of glaucoma progression was conducted based on parameters such as VFI (Visual field index), MD (Mean Deviation), PSD (Pattern standard deviation), and RNFL (Retinal Nerve Fiber Layer). As classification tools, the following algorithms were used: Multilayer Perceptron, Random Forest, Random Tree, C4.5, k-Nearest Neighbors, Support Vector Machine, and Non-Nested Generalized Exemplars. The best results, with an accuracy of over 90%, were obtained with Multilayer Perceptron and Random Forest algorithms. The NNGE algorithm also proved very useful in creating a hierarchy of the input values according to their influence (weight) on the considered outputs. On the other hand, the decision tree algorithms gave us insight into the logic used in their classification, which is of practical importance in obtaining additional information regarding the rationale behind a certain rule or decision.

## 1. Introduction

Artificial intelligence (AI) is a field of research with spectacular development and direct applications in Medicine and, also, in Ophthalmology. AI tools are used in the diagnosis and prognosis of open-angle glaucoma (POAG) with an accuracy that can sometimes exceed human performance.

AI tools commonly used in ophthalmology are machine learning classifiers (MLC). They establish the classes in which objects that have common functional or physical features can be distributed. These algorithms learn to take input data (e.g., clinical parameters) and automatically make a prediction (e.g., the presence of pathology, the severity of glaucoma, etc.) through supervised or unsupervised training processes. In supervised training, algorithms are trained on a fully labeled data set (for example, the diagnosis of the disease is labeled). In unsupervised training, algorithms are trained with an unlabeled data set, i.e., having as exclusive inputs only the clinical parameters in an attempt to identify new patterns/trends. These algorithms are usually well suited for managing numerical data (e.g., cup-to-disk vertical ratio, intraocular pressure, or age).

Decision trees are represented by a tree-like structure in which, based on successive tests, from general to individual, a new instance is introduced into an existing class [1]. When the Decision Tree is analyzed, it was observed that each path from the root to one of its leaves can be interpreted as a simple “if-else” rule. The well-known C4.5 algorithm develops a tree that is based on all input attribute values and then finalizes the decision rules by pruning [2].

In Nearest-Neighbor (NN) algorithms, an instance is introduced in the same class as its nearest neighbor in feature space or as the majority’s class of a group of neighbors [3]. The kNN algorithm (NN of order k) is not difficult to use and does not require a long calculation time. Another advantage is that it does not make assumptions about the data. In order to have good accuracy, it requires good quality data. The most challenging aspect is finding the optimal value for k [2].

Bayesian induction computes the membership probabilities of an instance to every class of the domains, selecting the class with the highest probability [4]. Bayesian networks are an extension of the naïve Bayes classifiers, as they take into account the fact that the events of the real world are not usually independent [5].

Neural networks can also be used as classifiers [6], as well as Support Vector Machine (SVM). SVM has the advantage of benefiting from well-established mathematical models, which means that it can approximate any measurable or continuous function with the desired accuracy. This is guaranteed by the principle of structural risk minimization in computer learning theory which always converges to a global optimum, unlike other learning methods such as ANN, which is based on the principle of minimizing empirical risk. In the case of neural networks, determining the optimal number of parameters (intermediate layers, hidden neurons, activation functions, etc.) requires time-consuming calculations. SVM has the advantage of a small number of parameters to be selected, so finding the optimal parameters becomes a feasible task. It can therefore be stated that SVM has better generalization capabilities than artificial neural networks and generally provides better results [7]. Obviously, this is a general theory, but depending on the available data and the problem, neural networks can outperform support vector machines. There are a few approaches known in the literature about the application of machine-learning models in medicine.

In 2012, Michael H. Goldbaum et al. [8] compared their own algorithm—Progress of Patterns (POP)—meant to identify the glaucomatous progression of the visual field by tracking classic visual field progression indices: VFI (Visual Field Index), MD (Mean deviation), and GPA (Guided Progression Analysis). This study shows that the POP model can detect progression with a high degree of confidence and can help the clinician to monitor glaucoma [8].

If in the previous article MLC was used in determining the progression of VF, in the study [9], Barella et al. worked with MLC using retinal nerve fiber layer (RNFL) data and optic nerve parameters obtained with spectral coherence optical tomography (SD-OCT) to investigate the diagnostic accuracy of these algorithms.

Ten MLC-type algorithms were tested: bagging (BAG), naïve Bayes (NB), linear support vector machine (SVML), Gaussian support vector machine (SVMG), multilayer perceptron (MLP), radial basis function (RBF), random forest (RAN), ensemble selections (ENS), classification tree (CTREE), and AdaBoost M1 (ADA). Weka software version 3.7.7 [10] (Waikato Environment for Knowledge Analysis, University of Waikato, Hamilton, New Zealand) was used to test all 10 classifiers. Both the characteristic operating curves of the receiver (ROC) and the calculation of the area under the ROC curve (aROC) were obtained using this software product. It was concluded that MLCs had good accuracy but did not improve the sensitivity and specificity of SD-OCT for the diagnosis of glaucoma [9].

Bizios et al. [11] aimed to conduct a study comparing MLC-type algorithms: artificial neural networks (ANN) and support vector machines (SVM). Measurements of the thickness of the retinal nerve fiber layer obtained using OCT were used as input data for the diagnosis of glaucoma. It was concluded that both classification algorithms worked very well, with similar diagnostic performances, and the input parameters have a greater impact on the diagnostic performance than the type of classifier used.

Wu et al. conducted a study [12] on 470 eyes from 265 patients, which were divided into normal patients and patients with glaucoma, and they were further divided into early, moderate and severe stages. All participants were examined with the Spectralis OCT, and other clinical parameters such as Visual Field mean deviation, IOP and visual acuity. In total, 114 parameters were used to train five machine learning algorithms, namely Conditional Inference Trees (CIT), Logistic Model Tree (LMT), C5.0 Decision Tree, Random Forest (RF), and Extreme Gradient Boosting (XGBoost). Random Forest had the highest accuracy (0.9327).

In the current paper, different classification algorithms are used in order to predict the evolution of glaucoma. The novelty of the methodology developed and presented in this article should be highlighted, which includes creating our own datasets from clinical observations from two different medical units, setting out different problems, respectively, choosing glaucoma-specific entry–exit groups, testing different classification algorithms, which have not been used on this type of problems before. Each algorithm can provide information on which measurements and values have a significant impact on the progression of glaucoma.

### Glaucoma

Glaucoma is an incurable condition that currently mainly affects the elderly and requires treatment and monitoring throughout the life of the patient, being the second leading cause of irreversible blindness worldwide. Population studies clearly show that Primary Open Angle Glaucoma (POAG) is a public health problem with increasing prevalence. The prevalence of glaucoma in the world in 2010 was 60.5 million; in 2020, the number reached 79.6 million (of which 60 million POAG); the prevalence in western countries is 30–50%. Approximately 6 million people have bilateral blindness due to glaucoma [13]. It is estimated that in 2040 approximately 111.82 million people will have glaucoma (79.76 million POAG and 32.04 million PCAG) [14]. Normal tension glaucoma is more common in East Asian countries (80–90%). In general, half of the patients diagnosed with glaucoma do not know that they suffer from this condition because the disease has a long asymptomatic evolution. The increased prevalence of glaucoma, the risk of blindness, and the treatable potential of this disease require perfecting the screening strategies for the detection of the disease, in which AI tools have an important role.

The main problem with glaucoma is the irreversible damage to the retinal ganglion cell layer, which suffers programmed cell death (apoptosis) followed by vision loss [13]. While an early diagnosis could minimize the risk of permanent vision loss [15], nearly half of glaucoma patients are diagnosed late due to the slow and asymptomatic course of the disease [16]. Once diagnosed, in order to predict the progression of glaucoma, multiple explorations are required, which results in repeated visits and time-consuming resources [17]. In 2016, the World Glaucoma Association acknowledged the absence of a single specific test that could predict glaucoma progression [18], demonstrating that some cases may be over- or under-treated. Because the changes in the retinal nerve fiber layer are irreversible, functional and structural assessment in a timely manner could help in early diagnosis and in better prediction of the glaucoma progression [19].

Major risk factors are increased IOP, advanced age, African American ethnicity, family history, and extensive excavation of the optic disc. Factors which are associated with an increased prevalence of the disease include myopia, diabetes, and altered ocular perfusion pressure. Altered vascular factors (blood pressure, vascular tone, infusion pressure) may also play an important role in the development and progression of glaucoma.

In patients with untreated POAG followed for 4 years, progression was detected in 44% of cases; structural progression (SD OCT macular and RNFL) is identified in 39% of cases and functional progression in 5% of cases, this being considered a low progression rate [13].

Risk factors for POAG progression are disc hemorrhage, increased mean IOP, IOP fluctuations, higher C/D ratio, and higher values of PSD at baseline [13].

Structural damage is monitored with the help of the OCT, which is able to identify changes (localized or diffuse) in the retina and detect RNFL thinning that precedes visual field progression. The structural progression consists of the extension of the pre-existing area, the deepening of the pre-existing area and the appearance of a new localized deficit. The progression of glaucoma is based on the linear regression of the retinal nerve layer in relation to age.

In recent years, a number of artificial intelligence (AI) tools have been used for the automatic segmentation and enhancement of ocular images in optical coherence tomography and for the processing of fundus images. Modern AI algorithms are specially adapted to extract significant features from complex and high-dimensional data for glaucoma screening, diagnosis, management and tracing based on the interpretation of functional and/or structural information [20].

Artificial intelligence can be used for early diagnosis, follow-up of patients, and assessment of the progression and prognosis of glaucoma. Thus, AI was used to interpret the photos of the fundus, OCT and VF. Although AI has a huge potential to revolutionize future glaucoma care, a number of challenges need to be overcome, such as the quantity and quality of data. Potential solutions to these challenges involve large-scale and diversified international collaborations for the collection of health data, tools to improve the quality of the data collection process, automatic integration of data from electronic medical records and regulations to ensure the security of not only the personal data but also of the analytical models [21].

## 2. Materials and Methods

### 2.1. Datasets

To predict the evolution of glaucoma, a dataset has been created containing information about glaucoma patients, examined in the Ophthalmology Department at the Countess of Chester Hospital in the United Kingdom. The study is retrospective and has analyzed the records of patients who had at least 3 follow-up visits during 2018–2021, in which all the required investigations were conducted.

The inclusion criteria in this study were: open-angle glaucoma or ocular hypertension with or without associated diabetes. Only patients in whom the following investigations were recorded at each of the three visits were included: visual field assessment, macular and optic nerve OCT, IOP measurement on both eyes, visual acuity measurement, pachymetry, HbA1c measurement, and documentation of baseline IOP at the time of glaucoma diagnosis. The criteria for the diagnosis of primary open-angle glaucoma (POAG) considered were: age greater than 40 years, IOP greater than 21 mm Hg without treatment at the time of diagnosis, open angles at gonioscopy, glaucomatous optic nerve (C/D ratio > 0.5), abnormal visual field (with Humphrey Field Analyzer perimeter) and thinning of retinal nerve fiber layers measured by optical coherence tomography-Heidelberg OCT. Patients with intraocular hypertension (OHT) were also included, in which it was used as a diagnostic criterion: age over 40 years, IOP greater than 21 mm Hg without treatment at the time of diagnosis, and open angles at gonioscopy. These patients may have a normal visual field, normal appearance of the optic nerve (C/D ratio < 0.5) and normal thickness of the retinal nerve layers measured on the OCT. In all patients, the corneal thickness was measured using the DGH Pachmate pachymeter.

The exclusion criterion from the study was represented by other types of glaucoma: normal tension, pseudoexfoliative, pigmentary, steroid-induced, neovascular and primary closed or narrow-angle glaucoma, and patients with an uncertain diagnosis of glaucoma.

Patients who did not have visual field examination and OCT at each of all three visits were excluded. Assessment of changes in diabetic retinopathy in a few patients with associated diabetes mellitus was conducted by examining the fundus (FO), using Volk lenses on the dilated pupil.

The criteria for classifying changes in diabetic retinopathy (DR) according to the Early Treatment Diabetic Retinopathy Study (ETDRS) were no changes (no signs of DR), early non-proliferative diabetic retinopathy (presence of a single microaneurysm), mild form (microaneurysms, hemorrhage in 2–3 quadrants, venous dilation in one quadrant), severe (micro-aneurysms, hemorrhages in all quadrants, venous dilation in 2–3 quadrants), and proliferative diabetic retinopathy (retinal neovascularization on the disc and retina in different quadrants).

A dataset was built with 47 attributes containing the following data: the visit number and the time elapsed between follow-ups in months, age of the patient, sex, age of glaucoma in years, age of diabetes in years, HbA1C divided in four categories (0 below 40 mmol/mol, 1 between 41 and 60 mmol/mol, 2 between 61 and 80 mmol/mol, and 3 between 81 and 100 mmol/mol), type of diabetes coded in three categories (0—without diabetes, 1—type 1 diabetes, and 2—type 2 diabetes), the stage of diabetic retinopathy numerically coded in the 5 categories mentioned above according to ETDRS (0—without retinopathy, 1—incipient non-proliferative diabetic retinopathy, 2—mild form, 3—severe form, and 4—proliferative diabetic retinopathy), the baseline IOP on each eye at the time of diagnosis before treatment, an IOP measurement at the time of each visit, the presence of intraocular lens (IOL) on each eye, numerically encoded (0—absent eye and 1—present), visual field values for each eye containing the following characteristics: VFI (Visual Field Index) measured in percentage, MD deviation (mean deviation), measured in dB and PSD (pattern standard deviation) measured in dB, visual acuity, measured in decimal values, for each eye, cup/disc ratio (C/D), horizontal and vertical for each eye, measured based on the digital photo recorded at the OCT examination, calculated in decimal values, a nerve layer thickness of the retinal fibers (RNFL) measured on OCT in microns, an average central retinal thickness (CRT) measured on OCT in microns, a central corneal thickness (CCT) measured with a pachymeter in microns, anterior chamber angle assessed by gonioscopy numerically coded in four categories (1—very narrow angle, 2—narrow angle, 3—moderately open angle, and 4—very open angle), the number of drops used to control glaucoma for each eye at the time of each visit, the number of surgical or laser glaucoma procedures performed to reduce IOP on each eye by the time of that visit.

Fifty patients were entered in the dataset, with three controls each, resulting in 150 visits with 300 entries (for both eyes). These are the records or instances of the dataset. In this first version of the dataset, the information regarding the three visits was considered independently, not being linked to the same patient.

In the second stage of modeling, the dataset was reconfigured in order to assign the 3 visits to a single instance (patient). Thus, the 300 entries were transformed into 100 (both eyes for 50 patients), bringing the data from each visit into a single row and thus increasing the number of columns (number of attributes). Values such as intraocular pressure, visual acuity, HbA1C, the presence of IOL, visual field indices (MD, PSD, and VFI), RNFL value, CRT value, retinopathy stage, number of surgeries, number of drops were assigned 1, 2, 3 at the end, thus corresponding to each visit. In this way, the applied algorithms were forced to take into account the fact that the data belong to the same patient and to be considered dynamically, being able to answer the question of whether or not glaucoma has progressed.

A third approach involves the application of classification algorithms on a dataset that included patients with glaucoma and diabetes from a prospective study [22]. There were 52 patients (101 eyes/instances) examined in the Ophthalmology Department at “St. Spiridon” Hospital of Iași. In this dataset, patients with POAG with associated diabetes were included. Patients with other types of glaucoma (pseudoexfoliative, pigmentary, steroid-induced, neovascular, closed angle) and those without associated diabetes were excluded. The diagnostic criteria for POAG were age over 35 years, IOP greater than 21 mmHg without treatment, open angles at gonioscopy, glaucomatous optic nerve damage (C/D ratio > 0.5), abnormal visual field (with Humphrey Field Analyzer perimeter) and thinning of the retinal nerve fiber layer (Optic Coherence Tomography-Cirrus HD OCT Carl Zeiss Meditec) [22]. Changes in diabetic retinopathy were diagnosed using indirect ophthalmoscopy with Volk lenses and retinal photographs with the Zeiss FO camera on a dilated pupil, analyzing all four quadrants. ETDRS criteria for the classification of diabetic retinopathy were used: no changes (absence of DR), a mild form of non-proliferative diabetic retinopathy (presence of a single microaneurysm), moderate form (microaneurysms, hemorrhages in quadrants 2–3, venous dilatations and soft exudates in one quadrant), severe form (microaneurysms, hemorrhages in all quadrants, venous dilation in 2–3 quadrants) and proliferative diabetic retinopathy (neovascularization of the disc and retina in different quadrants). Any new clinical feature that appeared at subsequent controls and led to the inclusion in severe non-proliferative or proliferative forms of diabetic retinopathy was considered to be the result of the progression of diabetic retinopathy [23].

The following values were selected as input parameters: glaucoma age (A), diabetes age (B), C/D ratio (C), glycosylated hemoglobin (D), intraocular pressure (E), patient age (F), MD—mean deviation (H) (MD is a parameter for measuring the degree of damage in glaucoma; the higher the negative value, the more important the change in glaucoma), and lens appearance (G). For the output parameter (I), the value “1” was used to mark the presence of DR changes and “0” for no changes.

To summarize, three datasets were used for the experiments, containing 300, 100, and 101 instances.

### 2.2. Modeling Methodology

In this paper, we attempt a comparison between several categorization methods, both sub-symbolic and symbolic. The algorithms used for solving the proposed problem are neural networks with different topologies, decision trees (the C4.5 algorithm, random tree, random forest), the Bayesian classifier, which tries to estimate the probability distribution of data, and an instance-based algorithm, which can be viewed as an extension of the classical nearest-neighbor approach. Each method can be applied with several variants by changing the parameters involved in the computing procedure.

Classification algorithms are well known in machine learning [24,25]. One of the purposes of these algorithms is to retain the nature of data distribution and to have good classifications with small prediction errors. A simple inductive learning structure is the decision tree. Given an instance of a concept specified by a set of properties, the tree returns a “yes” or “no” decision in relation to whether that instance belongs to a certain class. In order to avoid over-fitting, the resulting tree can be pruned at the end of the categorization process. In this way, the tree will be smaller, with more errors on the training set than the unpruned version, but supposedly with better generalization capabilities [26].

C4.5 is a decision tree induction algorithm that solves some problems such as overfitting data, treating continuous attributes and attributes with missing values, and increasing computational efficiency [27]. The C4.5 algorithm generates a decision tree by recursively partitioning the data set through a “depth-first” strategy [28].

Random tree classifier (RT) [24] builds a tree that takes into account k random features on each node and does not achieve any simplification. Therefore, the error percentage on the training set is quite low. Random forest [29] is composed of several classification trees. For the classification of a new object, the input vector is tested on each of the forest trees, and they propose a classification, considered as a “vote” for the respective class. The forest chooses the classification with the most votes.

The Reduced error pruning tree (REPT) is a quick classifier that builds a decision tree using informational gain or variance reduction and simplifies it to reduce error.

Bayesian induction of rules is based on Bayes’ theorem on conditioned probabilities. In classification, one is interested in the membership probabilities of an instance to a class based on the attribute values of that instance. The Naïve Bayes Classifier (NB) calculates the conditional probabilities of the classes assuming that all attributes are independent. A Bayesian network (BN) is a graph associated with a set of probability tables. Nodes represent variables, and arcs represent causal relationships between variables; therefore, in a Bayesian network, a variable is connected only to the variables on which it depends.

The Nearest-Neighbor (NN) algorithm classifies a new instance in the same class as the closest stored instance in the attribute space. A straightforward extension is the k-NN, where k neighbors (instead of 1) are taken into account when determining the membership of an instance to a class. This class of algorithms also includes the method of non-nested generalized exemplars (NNGE). NNGE is a nearest-neighbor-like algorithm using non-nested generalized exemplars as a way to avoid all forms of overgeneralization by never allowing exemplars to nest or overlap. NNGE always tries to generalize new examples to their nearest neighbor of the same class, but if this is immediately impossible due to intervening negative examples, no generalization is performed. If a generalization later conflicts with a negative example, it is modified to maintain consistency.

In the experimental studies described as follow, the performances of different models were assessed both on the whole training set (to estimate the capability of the model to fit the data at all) and using cross-validation (in order to determine the generalization capability of the model). Cross-validation is a procedure commonly used when comparing various machine learning models. It assumes that the dataset is split into a number of groups of folds, usually ten. Then, one group of data is used as a testing set and the rest of the nine as the training set. This is repeated ten times, with all the ten groups successively acting as testing sets. The overall performance indicators are eventually aggregated from the statistics of the ten testing sets.

It is well known that the accuracy of the results depends on the available dataset, both qualitatively (the correctness of the determinations and the uniformity of their distribution in the studied field) and quantitatively (the size of the dataset). Equally important for the credibility of the predictions is the way the problem is formulated, i.e., the selection of inputs and outputs.

In the first stage of the modeling, the initial version of the dataset was used (called dataset 1), which did not take into account the fact that the 3 visits belonged to the same patients. Each visit was treated as a different entity, and 4 subproblems were created. Every problem had the same 9 inputs but a different output (Figure 1). Thus, every number in the Output column corresponds to a different problem.

Secondly, a modified version of the same dataset was used (called dataset 2), in which the 3 visits were assigned to the same patient, the problem created for predicting the progression of glaucoma involved 12 inputs and one output, as shown in Figure 2.

Thirdly, a different dataset (dataset 3) that included patients with glaucoma and diabetes was used, containing: a total number of instances of 101, 9 input attributes (age of glaucoma, age of diabetes, C/D ratio, glycosylated hemoglobin, intraocular pressure, patient age, MD, the presence or absence of IOL), and one output-the existence (or not) of diabetic retinopathy.

## 3. Results

### 3.1. Prediction of Glaucoma Evolution Using Dataset 1

In the first attempt, 4 problems were proposed and solved, as Figure 1 shows, using a dataset composed of 300 instances.

In the problem labeled “1′′ (as shown in Figure 1), the output considered to predict glaucoma evolution was VFI, this being predicted according to the inputs which were risk factors in glaucoma. The results, represented by the correlation coefficient at the training and cross-validation (CV) phases, are presented in Table 1. The closer the correlation coefficient is to 1, the better the results. The algorithms used were: SVM, kNN, Random Forest, Random Tree, C4.5, and NNGE.

For each algorithm, only the configuration with the best CV result was included in the table. For SVM, the selected configuration uses the Pearson Universal Kernel (PUK) and the value of the cost parameter *C* = 100. Other attempts for SVM used, e.g., a polynomial kernel with degree *d* = 2 and the same *C* value giving quite bad results (0.7150, 0.4606), where the pair contains the coefficient of correlation for training and CV, respectively. SVM with Radial Basis Function (RBF) kernel, exponent *γ* = 1 and the same value for *C* yielded better results: (0.9500, 0.7453), but inferior to those obtained with the PUK kernel. For kNN, the value of *k* = 1, i.e., one nearest neighbor, was obtained by cross-validation, where the instances were weighted inversely proportional to the distance from the query point. The instance weighting procedure has no effect when *k* = 1, but it is important since several values for *k* were tried in order to find out the most effective value. For the Random Forest, *n* = 100 trees were used for the results presented in the Table 1. When using *n* = 1000, the following results were obtained: (0.9865, 0.8467), slightly better for training and slightly worse for CV. In order to apply C4.5 and NNGE, which require symbolic classes, the regression by discretization methods was applied, i.e., the output values were discretized into five intervals. The pruned version of C4.5 was designed to increase its generalization capability to the detriment of the training set accuracy, but in our case, it did not yield better results. The NNGE algorithm has a small set of parameters (including the number of attempts at generalization *n_ag_*) and was used with the default values of the Weka software (e.g., *n_ag_* = 5). All algorithms have additional parameters, but we tried different values for the ones which usually have the most influence on the results.

Table 1 shows the algorithm used as well as the correlation coefficient for the training and CV. The training results are better than the CV results, the last involving predictions about data not included in the training set. The fact that very high values of the correlation for CV were not obtained shows that the trees/rules are very complicated, so there is no simple model to predict the output using the model’s inputs. An additional cause is related to the available data, especially from a numerical point of view.

The best CV result with a correlation of 0.85 is provided by the Random Forest algorithm, which implies the best generalization capability.

The NNGE algorithm also calculates the weights for each input in relation to the considered output. These results can be used to see which inputs are more relevant to the output (Table 2). In this case, the Baseline IOP has the greatest influence on the output VFI.

The problem labelled number “2” (Figure 1) has as inputs the risk factors in glaucoma, and the output is MD (mean deviation). The results for this problem are shown in Table 3 and Table 4—training and cross-validation results are shown along with the values of the weights from the NNGE algorithm, pointing out the influence of each input on the considered output. Random Forest provides the best results.

As for the results in Table 1, more parameter values were tried for the algorithms. For example, by using SVM with a polynomial kernel with degree *d* = 2 and *C* = 100, we obtained (0.7501, 0.5931), while by suing SVM with a RBF kernel, *γ* = 1 and *C* = 100, we obtained (0.9467, 0.7623). In Table 3, *k* = 1 again for kNN, obtained by cross-validation. Random Forest with *n* = 1000 trees yielded (0.9891, 0.8760). C4.5 gave the same results in the pruned and unpruned variants after output discretization into both three and five values. The NNGE results in the table are obtained with discretization into five intervals. For three intervals, lower coefficients of correlation of (0.8284, 0.6984) were obtained.

In this case, the algorithm determined that the most significant weight on the MD output was Baseline IOP (intraocular pressure at the time of diagnosis), where the following medical conclusion can be drawn: the higher the initial pressure, the more the visual field will be affected.

The problem labelled number “3” (Figure 1) uses as output the value of PSD, and the results are shown in Table 5 and Table 6.

As with the results in Table 1 and Table 3, we can mention several alternative results: SVM with the polynomial kernel (0.7265, 0.6024) and RBF kernel (0.9500, 0.7527), Random Forest with 1000 trees (0.9882, 0.8616). The results for C4.5 and NNGE are reported after discretization into three intervals.

When aiming to predict the PSD output, Random Forest is, again, the algorithm that gives better results for cross-validation, having a correlation of 0.862. The greatest influence had the input “CCT”, with a weight of 0.138, determined by the NNGE algorithm.

Problem number “4” aims to predict the RNFL parameter, and the results are presented in Table 7 and Table 8.

In this case, some additional results obtained are: SVM with RBF kernel (0.9711; 0.8287); Random Forest with 1000 trees (0.9918; 0.9011). The results for C4.5 and NNGE are reported after discretization into three intervals. One can notice that the parameter values tend to affect the results in a similar way as presented in Table 1, Table 3 and Table 5.

Random Forest had a correlation coefficient of 0.90, higher than the other algorithms tested and higher than the one obtained in the previous cases. It seems that Random Forest is the best-suited algorithm for this kind of data to predict glaucoma progression.

The greatest impact on RNFL (the thickness of the retinal nerve layer) was the age of glaucoma and IOP at the time of diagnosis, which naturally leads to the medical conclusion: the older glaucoma is and the higher the initial baseline pressure, the thinner the RNFL will be.

### 3.2. Prediction of Glaucoma Evolution Using Dataset 2

The second approach includes problems illustrated in Figure 2 involving a dataset composed of 100 instances. The first dataset (dataset 1) was reconfigured, the three visits being assigned to the same patient and thus resulting in dataset 2. Algorithms such as Random Tree, C4.5, NNGE, Random Forest, MLP, SVM, and kNN were applied using the dedicated Weka software. The results are presented in Table 9, detailing how many correctly classified instances were obtained.

The best result was obtained with the MLP model, which correctly classified 92 instances out of 100. The Weka program used provided as a model an MLP network (12:8:1), i.e., with a single intermediate layer containing 8 neurons.

The C4.5 algorithm results in a decision tree that reflects the logic of classification. Figure 3 was automatically generated by the program and can be interpreted as follows: the algorithm made a very simple sorting in the first phase creating the following rule: if VFI at the 3rd visit was over 96%, it was considered that there was no progression in glaucoma. This may make medical sense, as patients with a VFI over 96% on their third visit are unlikely to have developed glaucoma progression. The next sorting was based on the PSD value—the algorithm found that there is glaucoma progression for PSD values >2.03.

Another type of result provided by the Weka program for each algorithm is the confusion matrix. This is a way to compare the desired results and the predicted results for each class. The rows represent the actual class, and the columns represent the predicted class. The cells of the matrix show how many instances belong to each combination. For the MLP model, the confusion matrix is shown in Table 10. This can be interpreted as follows: 13 results corresponding to YES (glaucoma has progressed) are correct, and 4 are wrong; 79 answers NO (glaucoma did not progress) are correct, and 4 are wrong. It is a breakdown of the overall result seen in Table 9.

Similar to the C4.5 algorithm, the Random Forest Tree provides a decision tree shown in Figure 4 that contains the graph corresponding to the decisions performed by the algorithm.

At first glance, it can be seen that Random Tree has created many more rules (20) compared to C4.5 (2). It is observed that this algorithm also focused mainly on VFI (Visual Field Index), which is a very accurate indicator for the progression of glaucoma, and the first two rules make medical sense. The algorithm highlights that if the VFI at the first visit is less than 99.5 and at the third visit is greater than 96.5, then it can be deduced that glaucoma did not progress, thus correctly classifying 31 results. Apart from this, the other decisions the algorithm took do not make sense from a medical point of view. The explanation may be related to the fact that Random Tree is one of the algorithms with the lowest performance, which is reflected in the fact that the correctly classified instances were 74, and those misclassified were 26. One of the reasons for this low performance may be the small size of the dataset on which the training was conducted. Therefore, a decision tree algorithm should only be considered reliable if its accuracy is high. However, instead of their direct use for prediction, the main advantage of random trees is that they can be combined into random forests, which usually create very accurate classifier models.

The same classification algorithms mentioned above were used to make predictions about treatment changes in certain patients. A column was added to answer this question, assigning “YES” if more drops were needed to control glaucoma at any point during the three visits or if surgical maneuvers were needed to control the pressure, and with “NO” the cases that were stable. This was conducted with the purpose of seeing whether the algorithms could detect which patients needed some alteration in their glaucoma management (surgical or medical). Eleven inputs were used: base IOP, IOP1, IOP2, IOP3, VFI1, VFI2, VFI3, RNFL1, RNFL2, RNFL3, and CCT, with a single output-TRATMOD (YES or NO). Satisfactory results were obtained, as shown in Table 11.

For the results presented in Table 11, Random Forest used 100 trees, C4.5 used the pruned tree version, kNN used 1 neighbor, MLP used eight neurons in the hidden layer, SVM used the PUK kernel, and AdaBoost used the Decision Stump weak classifier with 10 iterations. Once again, the multiplayer perceptron provided the best results, managing to predict 86 instances correctly out of the 100. Of the two decision trees, Random Tree and C4.5, the latter had superior results (Figure 5).

Figure 5 shows that most decisions of the C4.5 algorithm are made based on pressure control, as expected. In the first phase, the algorithm decides to treat all pressures above 24 mm HG, which is very close to the value recommended by the glaucoma guidelines. It is observed that in some cases, the corneal thickness was also taken into account, and, in total, the intraocular pressure was taken into account in each decision, usually choosing to treat the increased values. There are also some medical discrepancies, which is to be expected, given that the misclassification rate is 21%.

### 3.3. Prediction of Diabetic Retinopathy Status in Patients with Glaucoma Using Dataset 3

While the measurements from datasets 1 and 2 are made at the Countess of Chester Hospital in the UK, dataset 3 belongs to “St. Spiridon” Hospital of Iași, Romania and includes patients with glaucoma and diabetes, the total number of instances being 101. A previous approach [22] on the same data set tried to highlight a relation between diabetes and glaucoma using artificial neural networks. Therefore, a characteristic of dataset 3 is the fact that patients have both diabetes and glaucoma, which is also a justification for considering the RD parameter as an output.

Table 12 and Figure 6 present the results provided by the C4.5 algorithm. The results obtained speak for themselves, demonstrating a very high accuracy of the algorithm: 95 of the 101 instances were classified correctly (94.06%), and only six instances were incorrectly classified (5.94%). The mean absolute error was 0.0583.

In the present example, it is observed that the C4.5 algorithm found that all patients with diabetes aged under 10 years do not have diabetic retinopathy, which can be medically explained as the ocular manifestations tend to appear later. Then, among patients with a diabetes age of over 10 years, the algorithm made various other decisions but again focused on the age of diabetes, this time focusing on those with the age of diabetes under or over 14 years. In the majority (94.06%), the classifications made were correct.

Table 13 shows the results obtained by applying other classification algorithms to the same problem (detecting diabetic retinopathy) in patients with glaucoma and diabetes.

With the exception of the SVM, it can be stated that the other algorithms provided very good results, correctly classifying the instances more than 97% of the time. The 99% percentage, almost all correctly classified instances, was obtained using Random Forest. The algorithm also provided the best results in the previous examples on different datasets. It can be argued that Random Forest is an efficient algorithm for predicting eye diseases.

For the Random Tree algorithm, the results are graphically highlighted in Figure 7.

The Random tree algorithm had a result of approximately 97% accuracy in its classification. In Figure 7, it can be seen that the algorithm initially selected the patients according to age, over and under 72.5 years, then it considered the value of MD, and finally the age of diabetes, the value of glycosylated hemoglobin and the age of glaucoma.

At first glance, these rules found by the algorithm do not necessarily make medical sense for the diagnosis of diabetic retinopathy as inputs such as diabetes age and glycosylated hemoglobin value are used at the base of the tree. However, the algorithm has a high percentage of correct classification. This shows that the algorithm is able to highlight rules and connections that the clinician does not think of, and the reason why they cannot be extrapolated in the real world is the small number of patients in the dataset. However, potentially on a very large dataset, these erroneous connections would self-rectify.

## 4. Discussion

For the prediction of glaucoma, the available data were organized into three datasets. The first two sets of data contain information about glaucoma patients examined in the Countess of Chester Hospital in the UK, which examined, for each patient, at least three follow-ups in the period 2018–2021. Additionally, these patients had all the necessary investigations conducted in order to be included in the study. Inclusion, exclusion and classification criteria have been clearly formulated. If, in the first approach of the modelling, the three visits were considered independently (dataset 1), in the following approach, the dataset was reconfigured with the purpose of assigning the three visits to the same patient (dataset 2).

Another set of data (dataset 3) included patients with glaucoma and diabetes examined in the Ophthalmology Department at “St. Spiridon” Hospital of Iași, Romania.

Glaucoma risk factors that were taken into account consisted of patient age, sex, age of glaucoma and diabetes, HbA1c, baseline IOP, current IOP, the presence of IOL, and central corneal thickness. The glaucoma status was evaluated by VFI, MD, PSD, RNFL, number of antiglaucoma drops, and number of surgeries. What was assigned as inputs, respectively outputs, varied and was decided depending on the problem that was formulated.

The use of classification algorithms such as Random Tree, C4.5, NNGE, Random Forest, MLP, SVM, and kNN was conducted with the help of the dedicated Weka software.

The most important aspect of this paper was the development of a working methodology based on classification algorithms to assess the glaucoma progression according to risk factors specific to this condition.

In general, the Random Forest algorithm provided the best results, reaching an accuracy of 90% and MLP—92%, both being implemented with the help of the dedicated Weka software. The results of a similar approach were presented at the Romanian Annual Ophthalmology Reunion Conference in 2021 [30] when the same dataset was used, but feedforward neural networks were developed using the trial-and-error method. The best result for the cross-validation phase was accuracy of 90%. We also have to take into account the difficulty and time required to develop neural network models through successive tests, which consist of trial and error compared to the current research in which the Weka software provides the best result.

As an observation for situations where the weights hierarchy of input parameters does not match the clinical observations, the explanation is related to the accuracy of the NNGE algorithm, unsatisfactory in some cases, and the fact that the dataset should be larger. However, the working methodology and the potential of the algorithm should be considered, both in terms of precision and for the information provided in relation to the progression of the disease.

In a previous paper [22], we set out to predict the diabetic retinopathy status using dataset 3, which included patients from “St. Spiridon” Hospital, Iași. The modeling was performed with the help of MLP and JEN type networks (Jordan Elman Network), developed by the method of successive tests. At that point, good results were obtained; however, those proved to be inferior to the Random Forest algorithm. At the same time, this method, trial and error, involves a considerable amount of time and effort due to the repetitive process of training and validating each model. Only very large neural networks have provided satisfactory results-MLP (8:24:16: 1), MLP (8:40:32: 1), JEN (8:16: 1), and JEN (8:32: 1). The use of the Weka software from the current approach that involved different classification algorithms is simple and fast and can be performed by the non-specialized user with the help of software’s guided user interface.

A comparison can also be made with the results obtained in [31] because the same dataset was used in order to solve the same problem. The method applied for the prediction of diabetic retinopathy included support vector machines, designed with an evolutionary algorithm, respectively the Differential Evolution algorithm. It is also a laborious method based on software created by specialist users, and the highest accuracy was 84%.

The multitude of attempts involving different algorithms on different variations of the datasets aimed to determine which algorithm is best suited for the analysis of data from medical records, in particular, data on patients with glaucoma (possibly also with diabetes). This is a difficult task as it involves interpreting data that often has a high degree of subjectivity and low uniformity. Of all the algorithms used, the best results were obtained by MLP and RF. Random Forest and MLP are both black box algorithms, which, although they had an increased accuracy for cross-validation, do not provide any insight into the reasoning used in sorting data and making decisions. In contrast, decision tree algorithms show exactly the rules according to which the sorting was performed, the doctor being able to assess whether the rules make medical sense or not. Out of these, the best result was the Random Tree algorithm, which correctly classified diabetic retinopathy in 98 instances out of 101, basing its decisions largely on the age of diabetes.

## 5. Conclusions

The most important result of this paper is the fact that it developed a modeling methodology (classification) for predicting the evolution of open-angle glaucoma based on a series of data from clinical observations (medical records).

The efficiency of the approach comes not only from the satisfactory accuracy of the results but also from the type of output that was obtained, such as the option of ranking the weights of input values according to their impact on the progression of glaucoma (Non-Nested Generalized Exemplar provides such information) or by graphically illustrated decision trees (C4.5 and Random Tree algorithms), which show how the algorithm performs the classification.

As a general conclusion, Random Forest and MLP algorithms had the best results (more than 90% accuracy) in predicting glaucoma parameters, and the Baseline IOP value had a strong influence on the predictions. The glaucoma progression prediction was based on the VFI and PSD values, both of which represent visual field parameters. The algorithms were implemented in Weka, a dedicated software which has an intuitive user interface facilitating access to non-specialised users.

Artificial Intelligence methods prove, yet again, to be of a real benefit to the medical profession, the predictions obtained providing additional information and insight on the causes of the disease, as well as its evolution and treatment.

## Figures and Tables

**Figure 1 healthcare-10-01831-f001:**
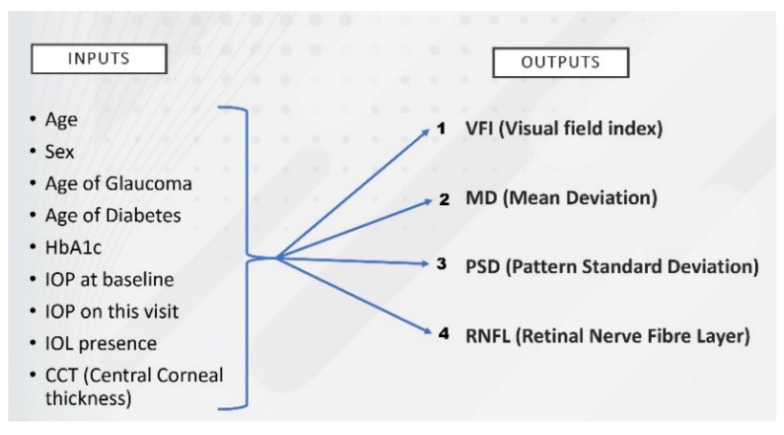
Inputs and outputs considered in the 4 problems created to predict glaucoma evolution.

**Figure 2 healthcare-10-01831-f002:**
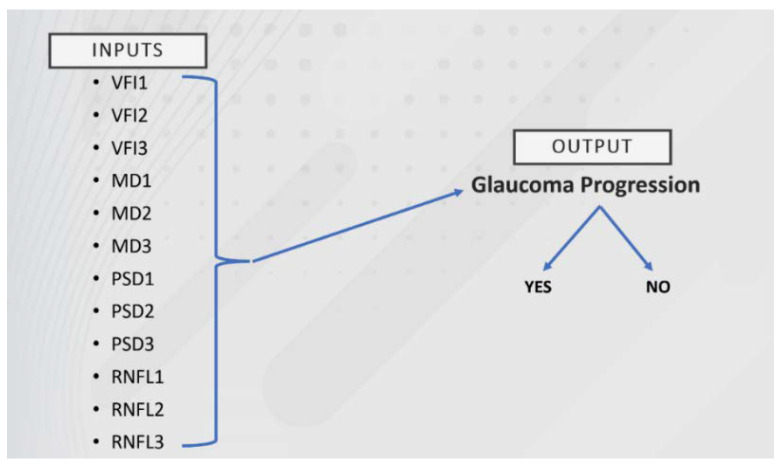
Selected inputs for predicting glaucoma progression for the modified dataset.

**Figure 3 healthcare-10-01831-f003:**
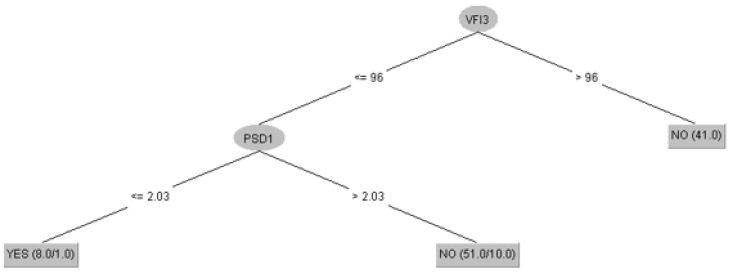
The decision tree generated by the C4.5 algorithm.

**Figure 4 healthcare-10-01831-f004:**
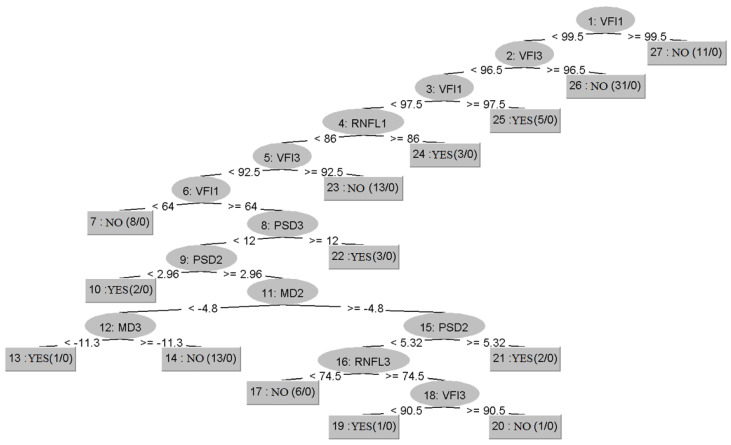
Decision tree generated by the Random Tree algorithm.

**Figure 5 healthcare-10-01831-f005:**
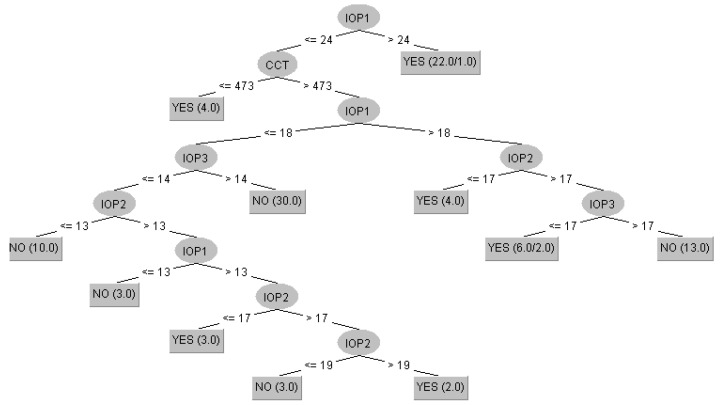
Decision tree of the C4.5 algorithm for predicting treatment change.

**Figure 6 healthcare-10-01831-f006:**
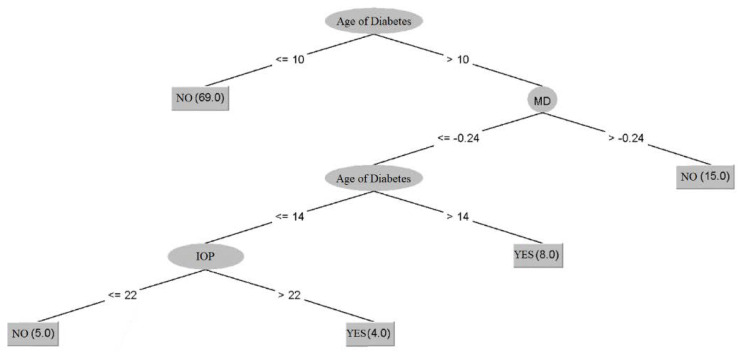
Results of C4.5 algorithm to predict diabetic retinopathy.

**Figure 7 healthcare-10-01831-f007:**
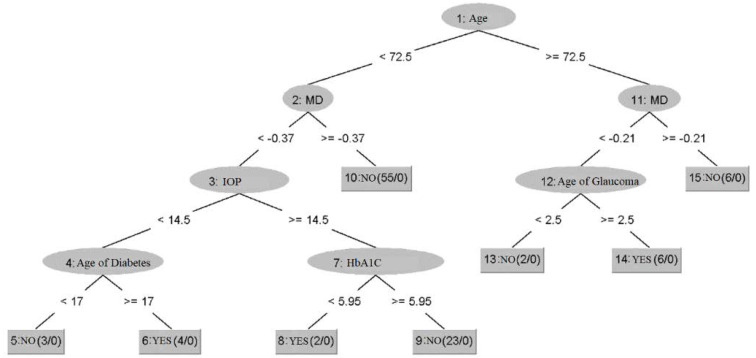
Results of the use of Random tree algorithm to predict diabetic retinopathy.

**Table 1 healthcare-10-01831-t001:** Results of classification algorithms to predict the VFI parameter based on glaucoma risk factors.

Algorithm	Training	Cross-Validation
SVM (PUK kernel, *C* = 100)	0.9990	0.7963
kNN (*k* = 1, *w_i_* = 1/*d_i_*)	0.9999	0.7168
Random Forest (*n* = 100)	0.9839	0.8558
C4.5 (unpruned)	0.8905	0.7279
NNGE	0.9558	0.6931

**Table 2 healthcare-10-01831-t002:** The weights for each input in relation to the VFI output calculated with the NNGE algorithm.

Inputs	Weights
Age	0.109
Sex	0.055
Glaucoma age	0.111
Diabetes age	0.088
HbAIc	0.066
Baseline IOP	0.150
IOP on this visit	0.072
IOL presence	0.0006
CCT	0.089

**Table 3 healthcare-10-01831-t003:** Results of classification algorithms to predict the MD parameter based on glaucoma risk factors.

Algorithm	Training	Cross-Validation
SVM (PUK kernel, *C* = 100)	0.9988	0.8039
kNN (*k* = 1, *w_i_* = 1/*d_i_*)	0.9999	0.7331
Random Forest (*n* = 100)	0.9882	0.8800
C4.5	0.9365	0.7252
NNGE	0.9630	0.7461

**Table 4 healthcare-10-01831-t004:** The weights for each input in relation to the MD output calculated with the NNGE algorithm.

Inputs	Weights
Age	0.127
Sex	0.062
Glaucoma age	0.109
Diabetes age	0.101
HbAIc	0.079
Baseline IOP	0.231
IOP on this visit	0.054
IOL presence	0.096
CCT	0.148

**Table 5 healthcare-10-01831-t005:** Results of classification algorithms to predict the PSD parameter based on glaucoma risk factors.

Algorithm	Training	Cross-Validation
SVM (*PUK kernel, C = 100*)	0.9957	0.7790
kNN (*k* = 3, *w_i_* = 1/*d_i_*)	0.9992	0.7893
Random Forest (*n* = 100)	0.9870	0.8620
C4.5 (pruned)	0.7962	0.6418
NNGE	0.9369	0.6842

**Table 6 healthcare-10-01831-t006:** The weights for each input in relation to the PSD output calculated with the NNGE algorithm.

Inputs	Weights
Age	0.086
Sex	0.027
Glaucoma age	0.079
Diabetes age	0.042
HbAIc	0.024
Baseline IOP	0.073
IOP on this visit	0.059
IOL presence	0.0009
CCT	0.138

**Table 7 healthcare-10-01831-t007:** Results of using classification algorithms to predict the RNFL parameter based on glaucoma risk factors.

Algorithm	Training	Cross-Validation
SVM (PUK kernel, *C* = 100)	0.9996	0.8184
kNN (*k* = 1, *w_i_* = 1/*d_i_*)	1.0000	0.8674
Random Forest (*n* = 100)	0.9896	0.9015
C4.5 (pruned)	0.8779	0.7724
NNGE	0.9144	0.6980

**Table 8 healthcare-10-01831-t008:** The weights for each input in relation to the RNFL output calculated with the NNGE algorithm.

Inputs	Weights
Age	0.092
Sex	0.015
Glaucoma age	0.142
Diabetes age	0.050
HbAIc	0.037
Baseline IOP	0.141
IOP on this visit	0.073
IOL presence	0.077
CCT	0.074

**Table 9 healthcare-10-01831-t009:** Results obtained with different classification algorithms for assessing glaucoma progression.

Algorithm	Correctly Classified Instances	Incorrectly Classified Instances	Kappa Statistic	Mean Absolute Error	Root Mean Squared Error
Random Tree	74	26	0.192	0.26	0.5099
Random Forest	85	15	0.3268	0.2116	0.3217
C4.5	83	17	0.1889	0.2006	0.3768
NNGE	81	19	0.1949	0.19	0.4359
kNN	86	14	0.4527	0.1478	0.3702
MLP	92	8	0.7165	0.1006	0.2724
SVM	83	17	0	0.17	0.4123

**Table 10 healthcare-10-01831-t010:** Confusion matrix for the MLP model.

		Predicted Class	
		YES	NO
Actual Class	YES	13	4
	NO	4	79

**Table 11 healthcare-10-01831-t011:** Results obtained with different classification algorithms for assessing the need for alteration of glaucoma treatment.

Algorithm	Correctly Classified Instances	Incorrectly Classified Instances	Kappa Statistic	Mean Absolute Error	Root Mean Squared Error
Random Tree	65	35	0.2757	0.35	0.5916
Random Forest	84	16	0.6497	0.2742	0.3541
C4.5	79	21	0.5329	0.2353	0.4285
NNGE	83	17	0.6298	0.17	0.4123
kNN	80	20	0.5379	0.2065	0.4425
MLP	86	14	0.7029	0.1436	0.3445
SVM	62	38	0	0.38	0.6164
AdaBoost	85	15	0.6664	0.2312	0.361

**Table 12 healthcare-10-01831-t012:** Performance indicators recorded by the C4.5 algorithm in the evaluation of diabetic retinopathy.

Performance Indicator	Value
Correctly classified instances	95 (94.0594%)
Incorrectly classified instances	6 (5.9406%)
Kappa statistic	0.7354
Mean absolute error	0.0583
Root mean squared error	0.2349

**Table 13 healthcare-10-01831-t013:** Results obtained with different classification algorithms for evaluating the occurrence of diabetic retinopathy.

Algorithm	Correctly Classified Instances	Incorrectly Classified Instances	Kappa Statistic	Mean Absolute Error	Root Mean Squared Error
Random Forest	100 99.0099%	1 0.9901%	0.9509	0.055	0.1094
SVM	88 87.1287%	13 12.8713%	−0.0186	0.1287	0.3588
MLP	97 96.0396%	4 3.9604%	0.8109	0.0558	0.1749
Random Tree	98 97.0297%	3 2.9703%	0.8631	0.0297	0.1723

## Data Availability

The simulation files/data used to support the findings of this study are available from the corresponding author upon request.

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
