# Peer review of "Classification Algorithms Used in Predicting Glaucoma Progression"

_healthcare, 2022, doi:10.3390/healthcare10101831_

Round 1
Reviewer 1 Report
1. Many papers are presented in the literature involving glaucoma diagnosis using ML and DL techniques. How does the proposed work stand apart from those?
2. How do the authors validate their model?
3. Comparison with other approaches can be provided in terms of datasets used, data splits taken, methods adopted etc.
4. Novelty of the work is not clearly stated.
5. Number of images tested, training : testing ratio need to be mentioned.
6. Organization of the content is strongly recommended.
7. Rigorous experimentation needed
Author Response
The responses are given point by point in the attached file.

Reviewer 2 Report
This is a well documented work that compares the efficiency of different classification algorithms in predicting glaucoma progression. The problem statement has been explained in detail in the manuscript. Previous work related to this field has been well cited and explained in the literature. However, below are the comments which can make the manuscript much better:
1. The specifications about the algorithms used for this study has not been explained in detail in the text. For example, what is the value of 'k' used for the KNN algorithm, the number of estimators in the random forest algorithm, etc. Such details could be included in the text with proper explanation (Table 1, 3, 5, 7, 10).
2. In Table 2 that showed the weights of different inputs in relation to the VFI input, why was the Glaucoma age considered to have the greatest influence while clearly comparing the values, it can be seen that the Baseline IOP has the highest weight value?
3. The confusion matrix shown in page 13 can be more clearly presented.
Author Response

(The authors gave the same response as above.)

Round 2
Reviewer 1 Report
Corrections are carried out
Author Response
The language was revised.
Thank you!